# Endothelial and Cardiovascular Effects of Naringin: A Systematic Review

**DOI:** 10.3390/nu17162658

**Published:** 2025-08-17

**Authors:** Jose A. Adams, Arkady Uryash, Alfredo Mijares, Jose Miguel Eltit, Jose R. Lopez

**Affiliations:** 1Division of Neonatology, Mount Sinai Medical Center, Miami, FL 33140, USA; auryash@msmc.com; 2Semyon & Janna Friedman Advanced Research Institute, Miami Beach, FL 33140, USA; lopezpadrino@icloud.com; 3Centro de Biofísica y Bioquímica, Instituto Venezolano de Investigaciones Científicas, Caracas 1020A, Venezuela; mijaresa@gmail.com; 4Department of Cellular, Molecular and Genetic Medicine, Virginia Commonwealth University, Richmond, VA 23298, USA; jose.eltit@vcuhealth.org

**Keywords:** naringin, flavonoid, endothelial function, myocardial protection, ischemia reperfusion, cardiovascular disease, oxidative stress, nutraceuticals

## Abstract

**Background/Objectives:** Naringin, a major flavonoid found in citrus fruits, has garnered significant attention over the past two decades for its potential cardiovascular benefits. This systematic review evaluates the effects of naringin on endothelial function and myocardial performance, with particular emphasis on ischemia-reperfusion (I/R) injury, based on the literature published from January 2000 to June 2025. **Methods:** The review was conducted in accordance with PRISMA 2020 guidelines. A comprehensive search of PubMed, Scopus, EMBASE, and Web of Science databases was performed using key terms including “naringin”, “cardiovascular”, “endothelial function”, “atherosclerosis”, and “ischemia-reperfusion.” A total of 62 studies were included and categorized into three domains: cellular models, animal studies, and human trials. Risk of bias assessments were conducted for each study type using appropriate tools. **Results:** Naringin consistently exhibited antioxidant, anti-inflammatory, and vasoprotective effects across all study types. Mechanistic studies highlighted the modulation of key signaling pathways, including PI3K/Akt, NF-κB, Nrf2, the renin-angiotensin system (RAS), and enhancement of KATP channel expression, as well as its ability to inhibit apoptosis, autophagy, and ferroptosis. In animal models, naringin improved endothelium-dependent vasorelaxation, reduced infarct size, and preserved myocardial function. Although limited, human trials reported beneficial effects on lipid profiles, arterial stiffness, and adiponectin levels. **Conclusions:** Naringin demonstrates strong potential as a dietary adjunct for cardiovascular protection, especially in the context of ischemic injury and vascular dysfunction. Further well-designed clinical trials are needed to define optimal dosing strategies and improve its bioavailability in humans.

## 1. Introduction

Cardiovascular diseases (CVDs) remain a leading cause of mortality worldwide, driven by factors such as atherosclerosis, hypertension, and myocardial infarction. Among the numerous dietary bioactives studied for cardioprotection, flavonoids have garnered significant attention due to their anti-inflammatory and antioxidant capacities. Naringin is a naturally occurring flavanone glycoside predominantly found in citrus fruits, particularly grapefruit (*Citrus paradisi*) and mandarin oranges (*Citrus reticulata*). It was first isolated in the mid-19th century by French chemist De Vry, who identified it as the compound responsible for the characteristic bitter taste of grapefruit juice [1,2]. Structurally, naringin is a flavanone glycoside composed of the aglycone naringenin linked to the disaccharide neo-hesperidose (Figure 1).

Naringin’s characteristic bitterness prompted early investigations in flavor and food chemistry. Although initially studied for its organoleptic properties, its first documented medicinal application dates back to the early 20th century, when it was explored for its antioxidant and capillary-stabilizing effects in traditional medicine. Following oral ingestion, naringin is metabolized primarily by gut microbiota or enzymes such as naringinase into its bioactive aglycone form, naringenin [4,5,6,7,8,9,10]. However, naringin exhibits very low oral bioavailability, typically less than 5%, due to its poor aqueous solubility, limited intestinal permeability, and extensive first-pass metabolism in the gut and liver [11]. To address these pharmacokinetic limitations, various advanced formulation strategies have been developed. These include solid dispersions, liposomes, nanosuspensions, nanoemulsions, and polymeric micelles—all of which aim to enhance solubility, absorption, and systemic exposure. For instance, liposomal encapsulation has been shown to significantly improve both the solubility and oral bioavailability of naringin in preclinical animal models [12].

Both naringin and naringenin have demonstrated a wide range of pharmacological effects, including antioxidant, anti-inflammatory, anti-apoptotic, and metabolic activities [1,2,3,4,5,6,7,8,9,10]. Over the past decades, interest in naringin has intensified due to its multifaceted biological actions and its potential protective role in cardiovascular health [9,13,14]. To date, there are 17 systematic reviews indexed under the search term “naringin”, with only one by Viswanatha et al. [8] specifically addressing its cardioprotective effects in preclinical in vivo and in vitro models. However, given that endothelial and myocardial function are central to cardiovascular health, a focused systematic evaluation of naringin’s effects on these domains is warranted. Our review uniquely addresses this gap by concentrating on endothelial and myocardial protection, with particular emphasis on ischemia-reperfusion injury, while also integrating available human clinical data and providing an in-depth analysis of underlying molecular mechanisms.

This systematic review summarizes evidence from January 2000 to June 2025 on the cardiovascular effects of naringin, focusing on its impact on endothelial function and myocardial ischemia (including ischemia reperfusion (I/R) injury). We examine findings from in vitro cellular studies, animal models, and human clinical studies, and we discuss the underlying mechanisms of action of naringin in the cardiovascular system. By compiling peer-reviewed data across these domains, we aim to provide a comprehensive overview of naringin’s cardioprotective potential and the molecular pathways involved.

## 2. Methods

### 2.1. Search Strategy

This review was conducted in accordance with the PRISMA 2020 guidelines. A comprehensive literature search was conducted using PubMed, Scopus, EMBASE, and Web of Science databases. The search was limited to peer-reviewed articles published between January 2000 and June 2025. Search terms included: “*naringin*” AND (“*cardiovascular*” *OR* “*endothelial function*” *OR* “*myocardial infarction*” *OR* “*ischemia-reperfusion*” *OR* “*hypertension*” *OR* “*cardioprotection*”). Boolean operators and MeSH terms were used where applicable.

### 2.2. Eligibility Criteria

Inclusion criteria were: (1) original experimental research; (2) peer-reviewed articles; (3) evaluation of naringin’s effects on endothelial or myocardial function; (4) in vitro, in vivo, or clinical study design; and (5) English language. Exclusion criteria included: reviews, abstracts, editorials, non-English texts, and studies lacking cardiovascular endpoints.

### 2.3. Study Selection

Two reviewers independently screened titles and abstracts of the retrieved records to identify studies relevant to naringin’s cardioprotective effects. Studies unrelated to cardiovascular or endothelial outcomes (e.g., those focusing on other pharmacological effects of naringin) were excluded at this stage. After the initial screening, full-text articles of the remaining references were obtained and assessed for eligibility based on the predefined inclusion criteria PICO Framework, incorporating Population/Model (**P**): Studies in cell cultures, animal models, or human subjects related to cardiovascular function or disease; Intervention(**I**): Exposure to naringin (isolated or as part of a formulation/diet); the Comparator (**C**) included placebo, untreated controls, or disease-only conditions; Outcomes(**O**): Cardiovascular or endothelial function outcomes (e.g., markers of endothelial function, myocardial infarct size, cardiac remodeling, blood pressure, etc.); and Study Type: Experimental studies (in vitro or in vivo), or clinical trials. Reviews, editorials, and irrelevant case reports were excluded.

A total of 2719 articles were excluded for irrelevance or not meeting criteria. Full texts of 165 articles were assessed, with 103 further excluded due to reviews (n = 40), irrelevant outcomes (n = 62), and a retracted article (n = 1). A total of 62 studies were included in this review.

### 2.4. Data Extraction and Synthesis

Data were extracted using a standardized form that included study type, model, dose, as well as duration of naringin treatment, endpoints assessed, and mechanistic findings. Due to heterogeneity in study design, endpoints, and model systems, a narrative synthesis approach was adopted. A qualitative synthesis approach was used to categorize results by model type (cell, animal, human) and primary endpoints (endothelial or myocardial function).

### 2.5. Risk of Bias Assessment

Risk of bias in the included studies was evaluated qualitatively by two independent investigators. In cases of disagreement, a third reviewer was consulted to reach consensus. The assessment considered factors such as study design, sample size, blinding, and reporting transparency. Studies exhibiting major methodological flaws were excluded. For human clinical trials, the Risk of Bias 2 (RoB2) tool was employed, while the SYRCLE risk of bias tool was used for animal studies [15]. A modified RoB2 approach, tailored for in vitro experiments, was applied to cellular studies. Risk of Bias tables, visualizations plot and summary were generated using the R package robvis, version 0.3.0. This tool supports domain-level visualization of RoB2 and generic risk of bias assessments [16].

### 2.6. PRISMA Statement

This review was conducted following the PRISMA 2020 guidelines. A PRISMA flow diagram summarizing the study selection process is found in Figure 2.

## 3. Results and Discussion

### 3.1. Cellular Studies on Endothelial Cells

A total of 28 in vitro studies utilizing various cell types were identified. Comprehensive details—including cell type, disease model, naringin concentration, treatment duration, and key findings—are provided in Appendix A. A modified SYRCLE-based risk of bias (RoB) assessment for cell culture studies is also included (Appendix A). The highest risk of bias was observed in domains related to cell authentication and blinding, while statistical reporting and the use of experimental replicates exhibited the lowest risk. Notably, eight of these studies specifically focused on endothelial cells. Naringin exhibits protective effects on vascular endothelial cells under various stress conditions. In cultured human endothelial cells, naringin significantly attenuates inflammatory activation and preserves normal endothelial function. For example, in human umbilical vein endothelial cells (HUVECs) stimulated with tumor necrosis factor-alpha (TNF-α), naringin dose-dependently inhibited the expression of cell adhesion molecules (VCAM-1, ICAM-1, E-selectin) and chemokines (CX3CL1/fractalkine, MCP-1, RANTES) that mediate leukocyte adhesion [10]. Mechanistically, naringin blocks TNF-α-induced activation of the nuclear factor kappa beta (NF-κB) pathway in these cells by suppressing the phosphorylation of IKKα/β, IκB-α, and NF-κB-p65, thereby preventing NF-κB nuclear translocation. This NF-κB inhibition explains the reduced inflammatory adhesion molecule levels and points to an anti-atherosclerotic effect of naringin at the endothelial level [17]. Similarly, naringin protected HUVECs from oxidative stress and inflammation triggered by other insults. In one study, naringin pretreatment prevented high-glucose-induced cell injury in human endothelial cells by scavenging reactive oxygen species (ROS) and down-regulating the chemokine CX3CL1, which is implicated in diabetic vasculopathy [18]. Another study showed that naringin counteracted TNF-α-induced oxidative damage in HUVECs by modulating multiple signaling pathways: it reduced NADPH oxidase (Nox4)–mediated reactive oxygen species (ROS) generation, inhibited NF-κB activation, and simultaneously activated the pro-survival PI3K/Akt pathway [19]. These changes translated into lower oxidative stress and inflammatory response in the endothelial cells. Furthermore, naringin protected endothelial cells from lipopolysaccharide (LPS) toxicity by dampening MAPK pathway activation, thereby reducing LPS-induced inflammation and apoptosis in vitro [20]. Collectively, these cell studies indicate that naringin preserves endothelial function by reducing oxidative stress, suppressing inflammatory signaling, and preventing endothelial apoptosis.

Naringin also influences endothelial cell behaviors related to vascular repair. Emerging evidence suggests a pro-angiogenic effect of naringin on endothelial cells under certain conditions. In vitro, naringin enhanced angiogenesis in HUVECs by up-regulating a specific PIWI-interacting RNA that promotes endothelial cell proliferation and tube formation [21]. Additionally, naringin was shown to protect endothelial cells against chemically induced injury. For instance, naringin prevented Trimethylamine N-Oxide (TMAO)-induced endothelial dysfunction in HUVECs by mitigating oxidative stress and inflammation caused by this pro-atherogenic metabolite [22]. Overall, across diverse in vitro models, naringin consistently improved endothelial cell viability and function, highlighting its direct cellular protective actions on the endothelium.

### 3.2. Cardiac Cells and Other Vascular Cells

Of the 28 cellular studies reviewed, 19 involved cardiovascular cell types, 14 focused on cardiac cells and five focused on vascular smooth muscle cells. Beyond the endothelium, naringin has shown beneficial effects on cadiomyocytes and vascular smooth muscle cells in vitro. In cardiomyoblast and cardiomyocyte models, naringin can blunt hypertrophic and apoptotic responses. A recent study in H9c2 rat cardiomyoblasts demonstrated that naringin markedly attenuates Angiotensin II–induced cardiomyocyte hypertrophy [23]. Naringin had the highest efficacy among tested flavonoids in that model, significantly reducing the Ang II–triggered increase in cell size and the expression of hypertrophy-related proteins. Notably, naringin’s anti-hypertrophic effect was linked to its ability to inhibit carbonic anhydrase II (CA-II) and downstream ion transporters. Naringin bound strongly to CA-II and dose-dependently inhibited its activity, which in turn suppressed the upregulation of the Na^+^/H^+^ exchanger-1 (NHE1) and Na^+^/Ca^2+^ exchanger (NCX1) in Ang II–treated cells. Since CA-II and NHE1 are key contributors to intracellular pH and Ca^2+^ overload in cardiac hypertrophy, their inhibition by naringin helps prevent the pathological remodeling process. In summary, naringin effectively prevented Ang II–mediated cellular hypertrophy, likely by targeting CA-II and related ion exchange pathways to normalize ionic homeostasis in cardiomyocytes [23]. The vasorelaxant effect of naringin on endothelium denuded vascular smooth muscle cells has been studied and shown in part to be mediated by activation of large conductance Ca^2+^ activated K^+^ (BK_Ca_) channel opener [24]. Recent evidence also suggests that naringin exerts an antiarrythmic effect in isolated cardiomyocytes by inhibiting late Na^+^ currents, peak Na^+^, Ca^2+^, and K^+^ currents in a concentration-dependent manner [25].

Using cellular models of high-glucose induced cardiomyocyte injuries, various investigators have shown that naringin attenuates glucose induced injury and inflammation via modulation of the Leptin-JAK2/STAT 3 pathway [26,27], upregulation of ATP-sensitive K^+^ (K_ATP_) channels, inhibition of the NF-κB pathway [28], and inhibition of the ROS activated mitogen-activated protein kinase (MAPK) signaling pathway [29].

Naringin has also been reported to protect cardiomyocytes from simulated ischemia-reperfusion in vitro. In a hypoxia/reoxygenation injury model using H9c2 cells, naringin (at non-cytotoxic concentrations) improved cell survival, reduced oxidative stress, and preserved mitochondrial membrane potential following hypoxic injury [30]. These cardioprotective effects in vitro were associated with the inhibition of the cyclic GMP-AMP Synthase, which stimulates interferon genes (cGAS-STING) pathway, an innate immune DNA-sensing pathway that can drive inflammation and cell death (including ferroptosis) after ischemic stress. By suppressing cGAS-STING activation in cardiomyocytes, naringin reduced downstream oxidative damage and ferroptotic cell death, thereby improving cardiomyocyte viability under I/R-like conditions [30]. This finding aligns with other evidence that naringin bolsters cellular antioxidant defenses (potentially through (nuclear factor erythroid 2-related factor 2) Nrf2 activation, as discussed later) and prevents regulated cell death in cardiac cells. Further, naringin protects cardiomyocytes against doxorubicin-induced cardiotoxicity with reduction in apoptosis, ROS generation, and attenuation of loss of matrix metallo porphyrins (MMP) via suppression of the expression and activity of p38MAPK, important for protection in chemotherapy induced myocardial injury [31].

In vascular smooth muscle cells (VSMCs), naringin exhibits anti-atherogenic effects by curbing abnormal proliferation and migration. For instance, naringin was shown to induce a cell-cycle arrest at the G-1 phase in cultured VSMCs by activating the Ras/Raf/ERK pathway to increase expression of the cyclin-dependent kinase inhibitor p21WAF1 [32]. Additionally, naringin inhibited TNF-α-induced matrix metalloproteinase-9 (MMP-9) expression in VSMCs via down-regulating Akt phosphorylation, which could stabilize atherosclerotic plaques by reducing extracellular matrix degradation [33]. Although these VSMC studies are not directly about endothelial or myocardial function, they reinforce naringin’s broad cardiovascular protective profile at the cellular level—spanning endothelial cells, cardiac myocytes, and smooth muscle cells.

In summary, in vitro studies consistently demonstrate that naringin favorably modulates key cellular processes in the cardiovascular system; it lowers oxidative stress and inflammation in endothelial cells, prevents cardiomyocyte hypertrophy and death, and inhibits pathological VSMC activation. These cellular actions provide mechanistic underpinnings for the benefits observed in whole-animal models of cardiovascular disease. Key outcomes for each of the cellular studies can be found in Appendix A.

### 3.3. Animal Models

A variety of animal studies have evaluated naringin’s effects on cardiovascular health, using models of endothelial dysfunction, atherosclerosis, hypertension, and myocardial ischemia. A total of 29 animal studies meeting the inclusion criteria were reviewed. Of these, 15 utilized models of metabolic disorders, including diabetes, hypercholesterolemia, and hyperlipidemia. Nine studies specifically addressed myocardial ischemia-reperfusion injury or hypertension. The SYRCLE-based Risk of Bias (RoB2) assessment is provided in Appendix A. The domains with the highest risk of bias were allocation concealment, random housing, caregiver blinding, and blinding of outcome assessors. Overall, these in vivo studies corroborate the protective actions of naringin suggested by cellular experiments, showing improvements in both endothelial function and cardiac function across multiple disease models.

#### 3.3.1. Atherosclerosis and Endothelial Dysfunction

Naringin has demonstrated anti-atherosclerotic effects in animal models of hyperlipidemia and endothelial injury. In cholesterol-fed rabbit models, chronic naringin supplementation reduced the development of atherosclerotic lesions. Two early studies in hypercholesterolemic rabbits found that naringin significantly attenuated aortic atherosclerosis, and this was associated with lower expression of vascular cell adhesion molecule ICAM-1 in the endothelium [34,35]. By down-regulating adhesion molecules, naringin likely reduced monocyte infiltration into the artery wall, thereby slowing plaque formation. Another rabbit study showed that naringin’s antioxidant properties contribute to its anti-atherogenic outcome: when high-cholesterol-fed rabbits were treated with naringin (alone or alongside low-dose lovastatin), there was a marked reduction in oxidative stress markers in the aorta and an improvement in plasma lipid profiles [36,37]. These results indicate naringin can mitigate the oxidative damage and dyslipidemia that drive atherosclerosis progression. Similarly, in apolipoprotein E-deficient mice (a model prone to atherosclerosis), naringin supplementation significantly inhibited plaque development, protected vascular endothelium by reducing oxidative stress via multiple metabolic pathways, and promoted endothelial derived nitric oxide (eNOS) protein expression [38]. The Chanet (2012) study specifically noted naringin slowed the progression of aortic lesions in diet-induced hypercholesterolemic mice, highlighting the flavonoid’s potential to interfere with atherogenic processes in vivo [38]. In a similar mouse model of dyslipidemia, Zhang et al. showed that naringin reduces oxidative stress and protects the vascular endothelium from dysfunction induced by dyslipidemia through multiple metabolic pathways [39]. A recent metabolomic and network pharmacology analysis suggested that the PI3-AKT pathway is an important mechanism for the effect of naringin on atherosclerosis. The data suggested that naringin likely inhibits the PI3K-AKT/TL4/NF-kB pathway to decrease inflammation during development of atherosclerosis [40]. In a Type 2 diabetes (db/db) mouse model, Uryash et al. showed aberrant diastolic Ca^2+^ increased pro-inflammatory proteins (TNF-α, NF-κB), decreased cell viability, and reduced expression of K_ATP_ channel and subunits Kir6.2, SUR1, SUR2, in cardiomyocytes. Naringin treatment reduced Ca^2+^ overload, improved cardiomyocyte glucose transport, reduced ROS production and restored protein expression of KATP and its subunits [41].

Naringin’s ability to improve endothelial function in vivo is a recurring finding. A notable study by Pengnet et al. (2019) [42] evaluated naringin in rats fed a high-cholesterol diet, which induces endothelial dysfunction and oxidative stress. After 8 weeks, rats that received naringin (100 mg/kg/day) had significantly better endothelium-dependent vasodilation in isolated aortic rings compared to untreated hypercholesterolemic controls. Specifically, naringin restored the blunted vasodilatory response to acetylcholine (ACh) observed in hypercholesterolemic rats, indicating an improvement in endothelial nitric oxide (NO) bioavailability [42]. Consistently, aortic NO levels (assessed by nitrate/nitrite content) were higher in the naringin-treated group, and vascular superoxide levels were lower, reflecting reduced oxidative quenching of NO. Mechanistic analysis of aortic tissue showed that naringin sharply down-regulated the protein expression of LOX-1 (lectin-like oxidized LDL receptor-1) as well as key subunits of NADPH oxidase (including Nox2, Nox4, and p47^phox^) that produce superoxide radicals. Naringin also reduced the overexpression of inducible nitric oxide synthase (iNOS) and markers of nitro-oxidative damage (like 3-nitrotyrosine and 4-HNE adducts) in the aortae of these rats. By down-regulating LOX-1 and NADPH oxidase, naringin likely curbed the production of ROS that cause endothelial dysfunction, thereby preserving NO and normal endothelium-dependent relaxation [42]. Using a rat model of high-fat (diet-induced) obesity and cardiovascular dysfunction, Alam et al. showed that 8 weeks of oral naringin reduced plasma lipids, improved oxidative stress markers, inflammatory markers, and improved mitochondrial dysfunction. Physiologically, naringin normalized systolic hypertension and improved vascular and ventricular diastolic function [43]. In another diet-induced (fructose-fed) metabolic syndrome rat model, Malakul et al. showed that 4 weeks of oral naringin significantly reduced endothelium dysfunction and restored vascular eNOS and p-eNOS protein expression [44]. Collectively, these findings provide direct in vivo evidence that naringin improves endothelial function in the setting of hypercholesterolemia and metabolic syndrome by reducing oxidative stress and inflammatory signaling in vessels and restoring NO bioavailability through vascular eNOS expression.

#### 3.3.2. Hypertension and Cardiac Hypertrophy

In models of hypertension, naringin has shown antihypertensive effects and prevented cardiac remodeling. An example is the L-NAME induced hypertension model, where chronic nitric oxide synthesis inhibition leads to high blood pressure, endothelial dysfunction, and left ventricular (LV) hypertrophy. Khamseekaew et al. treated L-NAME hypertensive rats with naringin (20 or 40 mg/kg/day), effectively normalizing blood pressure and preserving cardiac function comparable to standard therapy [4]. Systolic blood pressure, which was significantly elevated by L-NAME, remained near normal in naringin-treated rats (similar to the effects of the antihypertensive drug telmisartan). Consequently, naringin-treated rats did not develop the LV ejection fraction and fractional shortening decline seen in hypertensive controls. Naringin also protected both the aorta and heart from structural remodeling: untreated L-NAME rats showed pronounced LV hypertrophy, interstitial fibrosis, and aortic wall thickening, whereas those changes were absent or mild in naringin-treated rats. Importantly, naringin prevented the impairment of endothelium-dependent aortic relaxation that was observed in the hypertensive animals, indicating that it preserved endothelial function even under the NOS-inhibited, high-pressure conditions. The study revealed that naringin blunted the activation of the renin-angiotensin system (RAS) and oxidative pathways in hypertension: levels of circulating Ang II and downstream mediators (e.g., AT1 receptor, PKC, and NADPH oxidase subunit NOX2 in cardiac tissue) were significantly lower in naringin-treated rats than in hypertensive controls. Naringin also reduced the hypertension-associated increase in pro-inflammatory cytokine TNF-α and markers of oxidative stress in plasma. Through these actions—reducing RAS overactivity, oxidative stress, and inflammation—naringin alleviated the hemodynamic burden and prevented organ damage in hypertensive rats [4]. These results suggest naringin could be a promising adjunct for managing hypertension, as it addressed both the cause (RAS-mediated vasoconstriction) and consequences (endothelial dysfunction and cardiac remodeling) of high blood pressure.

Consistent antihypertensive and vasoprotective effects of naringin have been noted in other studies. Ikemura et al. (2012) [45] reported that naringin supplementation prevented the onset of hypertension and reduced the incidence of stroke in stroke-prone spontaneously hypertensive rats. Naringin-treated SHR had lower blood pressure and were protected from cerebrovascular thrombotic events compared to controls, presumably due to naringin’s vascular benefits (the study also tested hesperidin with similar findings) [45]. In diet-induced metabolic syndrome models, where rodents develop hypertension alongside obesity, naringin likewise helped prevent blood pressure elevation. For example, in rats fed a high-carbohydrate, high-fat diet, naringin (~100 mg/kg/day) averted the rise in systolic blood pressure that occurred in untreated obese rats [43]. This occurred in parallel with improvements in insulin sensitivity and lipid levels, indicating naringin’s multifaceted benefits in metabolic syndrome. Other investigators in similar models of diet induced metabolic syndrome have shown that naringin decreased cardiac hypertrophy and decreases apoptosis via ROS dependent ATM (Ataxia-telangiectasia mutated protein kinase)-mediated p53 signaling [46]. Collectively, these animal results establish that naringin can lower blood pressure and improve vascular reactivity in both genetic and acquired models of hypertension as well as structural remodeling, thereby protecting the heart and vessels from hypertensive injury.

#### 3.3.3. Myocardial Ischemia and Infarction

Some preclinical studies have examined naringin’s effects on myocardial infarction and ischemia-reperfusion (I/R) injury––conditions where cardiomyocyte death and dysfunction occur due to ischemia and oxidative stress. Naringin has consistently shown cardioprotective effects in these models. In a rat model of acute myocardial I/R injury (induced by coronary artery occlusion followed by reperfusion), naringin pretreatment significantly reduced myocardial damage and improved cardiac function [47]. Specifically, rats given naringin (typically 50–100 mg/kg, intravenously or orally before ischemia) exhibited smaller infarct sizes after I/R compared to untreated rats. Biochemical markers of myocardial injury, such as creatine kinase-MB (CK-MB), lactate dehydrogenase (LDH), and cardiac troponin I, were all markedly lower in naringin-treated animals, indicating less cardiac cell necrosis during I/R. Li et al. (2021) [47] also found that administering naringin at 100 mg/kg to rats prior to ischemia led to significantly reduced release of CK-MB, LDH, and troponin following 30 min of coronary occlusion and 3 h of reperfusion. Correspondingly, histological examination showed that naringin preserved myocardial tissue architecture, with less hemorrhage and necrosis than in controls [47]. Importantly, cardiac functional recovery was improved by naringin: echocardiography after I/R revealed that naringin-treated rats had higher left ventricular ejection fraction (EF) and fractional shortening (FS), and smaller end-systolic diameter, than untreated I/R rats. Naringin helped maintain cardiac contractile function post-I/R, whereas control hearts showed significant systolic dysfunction (low EF/FS) due to infarction. Hemodynamically, naringin also stabilized parameters like mean arterial pressure during reperfusion in some studies, suggesting it mitigates I/R-induced pump failure [47]. These findings have been reproduced across different laboratories, strongly indicating that naringin exerts a direct infarct-sparing and cardioprotective effect in acute myocardial ischemia models [48,49,50].

The cardioprotection by naringin in I/R is accompanied by reductions in myocardial apoptosis, oxidative stress, and inflammation. For instance, TUNEL assays and caspase-3 measurements in I/R hearts show that naringin markedly lessens cardiomyocyte apoptosis compared to untreated I/R injury [47]. Pro-apoptotic markers like cleaved caspase-3 are decreased, while anti-apoptotic Bcl-2 is preserved with naringin, reflecting its anti-apoptotic influence on the myocardium [47]. Inflammatory cytokines that surge during reperfusion (TNF-α, IL-1β, IL-6) are significantly lower in naringin-treated hearts [47]. This study also noted that myocardial levels of TNF-α, IL-1β, and IL-6 were all elevated in I/R control rats (indicative of an acute inflammatory response), whereas naringin pretreatment inhibited the release of these cytokines by the reperfused heart tissue. Similarly, oxidative stress markers respond favorably to naringin: malondialdehyde (MDA, a lipid peroxidation product) was reduced and endogenous antioxidant enzyme activity (e.g., superoxide dismutase, SOD) was higher in naringin-treated I/R hearts than in controls [47]. These changes imply that naringin helps maintain redox balance in the heart during I/R, likely by bolstering antioxidant defenses and scavenging free radicals generated upon reperfusion.

Mechanistic studies in I/R models have begun to pinpoint the pathways through which naringin confers cardioprotection. A recurring theme is the activation of the PI3K/Akt signaling pathway. In the rat I/R study by Li et al., naringin’s beneficial effects were largely abrogated by co-administration of LY294002 (a PI3K/Akt inhibitor), suggesting that Akt activation is required for naringin’s full cardioprotective action [47]. In naringin-treated hearts, phosphorylated Akt levels were significantly increased, which is known to trigger downstream survival pathways and inhibit cell death mechanisms. One consequence of Akt activation by naringin was the suppression of excessive autophagy in the reperfused heart: naringin reduced the upregulation of autophagy markers (like Beclin-1 and the LC3B-II/LC3-I ratio) that typically occurs during I/R, thereby preventing autophagy-mediated cell death. By contrast, blocking Akt with an inhibitor partly reversed these effects, leading to higher Beclin-1 and LC3-II levels despite naringin, and diminished the functional protection [40]. These findings indicate that PI3K/Akt-dependent signaling is a key mediator of naringin’s ability to reduce myocardial injury and dysfunction in I/R settings. Additional mechanisms from other studies include the modulation of iron-dependent necrosis (ferroptosis) via inflammatory pathways: Zhang et al. (2025) [35] showed that naringin inhibited the cGAS-STING pathway in a myocardial I/R model, thereby reducing downstream inflammatory injury and ferroptosis in the heart. STING is an immune sensor that exacerbates I/R damage by driving cytokine release and cell death; its inhibition by naringin further explains the lower inflammation and infarct size observed with treatment [30]. Other investigators using models of myocardial ischemia reperfusion in rats have also shown the effects of infarct reduction are linked to the nuclear factor-erythroid factor 2 (Nrf2 and glutathione peroxidase 4 (GPX4) axis [49], and reduction in endoplasmic reticulum stress via cyclic guanosine monophosphate(cGMP)-depending protein kinase (PKG) signaling [50].

Beyond acute I/R, naringin has shown benefits in other forms of myocardial injury. In isoproterenol-induced myocardial infarction in rats (a non-ischemic model of cardiac oxidative injury), naringin pretreatment was reported to prevent oxidative stress and myocardial necrosis, as evidenced by preserved antioxidant enzyme levels and reduced pathological changes in the heart [51]. In diabetic cardiomyopathy models (e.g., streptozotocin-induced diabetic), naringin improved cardiac histology and function, likely through its glycemic control and antioxidant effects (though in some cases, naringin primarily improved lipid profiles and inflammation without fully normalizing hyperglycemia [52,53,54]. Naringin also attenuated sepsis and lipopolysaccharide (LPS) induced myocardial dysfunction, and oxidative stress [55,56,57]. In yet another model of metabolic induced cardiac dysfunction (sweetened alcohol) in female rats, naringin attenuated the concentric remodeling of the heart [58]. Using a chemotherapeutic (Doxorubicin)-induced myocardial injury mouse model, Zhao et al. showed that administration of oral naringin for 15 days was able to attenuate myocardial injury with reduction in apoptosis, inflammation, and oxidative stress [59]. While each model has unique pathophysiology, a common observation is that naringin limits myocardial cell injury (from infarction, metabolic stressor, inflammation, or toxicity) by reducing oxidative damage, restraining inflammatory and stress signaling, and promoting cell-survival pathways in the heart. Key outcomes for each of the animal studies can be found in Appendix A.

In summary, animal studies robustly support naringin’s cardioprotective role. Naringin improves endothelial function in vivo (enhancing vasodilation and reducing atherogenic endothelial activation) and protects the myocardium in vivo (preventing hypertrophy, reducing infarct size, and preserving cardiac function after injury). These benefits have been replicated in models of atherosclerosis, hypertension, myocardial infarction, and heart failure. The breadth of positive outcomes across different species and disease contexts underscores naringin’s potential as a broad-spectrum cardiovascular protective agent.

### 3.4. Human Studies

Five human clinical trials satisfied the inclusion criteria for this review. The RoB2 assessment revealed a low risk of bias in most domains (Figure 3).

A comprehensive summary, including study type, population, naringin dosage, treatment duration, as well as key outcome measures, is shown in Table 1.

Evidence for naringin’s cardiovascular effects in humans, while still limited, has begun to emerge from clinical trials and dietary intervention studies. Compared to the wealth of preclinical data, relatively few human studies have isolated the impact of naringin (as a supplement or dietary component) on cardiovascular endpoints. Nonetheless, the available trials indicate that naringin may confer metabolic and vascular benefits in people, aligning with the trends seen in animal models.

#### 3.4.1. Metabolic and Lipid Profile Improvements

A double-blind randomized controlled trial by Barajas-Vega et al. (2022) investigated naringin supplementation in human subjects with dyslipidemia [64]. In this trial, 28 adults with elevated cholesterol were randomized to receive either naringin 450 mg once daily or a placebo, for a period of 90 days. The results showed significant improvements in several cardiometabolic parameters in the naringin group compared to placebo. Body mass index (BMI) modestly decreased with naringin (from ~33.3 down to 30.6 kg/m^2^ on average), whereas it remained higher in the placebo group (no improvement, ~33.3 kg/m^2^). More strikingly, naringin produced a significant reduction in serum total cholesterol and low-density lipoprotein (LDL) cholesterol levels: by 3 months, mean total cholesterol was ~182 mg/dL in the naringin group versus 245 mg/dL in placebo (a ~25% reduction, *p* < 0.01), and LDL was ~100 mg/dL with naringin versus 125 mg/dL with placebo (*p* = 0.03). These changes indicate a favorable lipid-modulating effect of naringin in humans, consistent with its hypolipidemic effects noted in animal studies. Additionally, the trial measured adiponectin, an anti-inflammatory adipokine inversely related to cardiometabolic risk. Adiponectin levels rose significantly in the naringin-treated patients (0.82 μg/mL vs. 0.59 μg/mL in placebo, *p* = 0.01), suggesting an improvement in insulin sensitivity and antiatherogenic profile. Recently, Lopez-Almada et al. (2023) [65] have extensively reviewed the effects of naringin as a regulator of obesity and appetite providing compelling and plausible explanation for the observed modest decrease in BMI as the effects of naringin on modulation of peptides directly associated with the hunger–satiety pathway, such as ghrelin, cholecystokinin, insulin, adiponectin, and leptin. The authors concluded that naringin could be a useful adjunct in managing components of metabolic syndrome, given its ability to reduce weight and improve the lipid profile and adipokine balance. This controlled trial provides clinical evidence that the metabolic benefits of naringin seen in rodents translate to humans: improved cholesterol handling and potentially reduced obesity-related risk.

It is worth noting, however, that human outcomes with naringin have not been universally positive, possibly due to differences in dose and bioavailability. For instance, an earlier dietary intervention study in moderately hypercholesterolemic adults found that naringin 500 mg/day for 8 weeks did not significantly change plasma cholesterol levels compared to baseline [61]. The lack of effect in that case may be due to the dose being insufficient or the study duration too short; as suggested by Alam et al. (2013), translating effective rodent doses to humans might require around 1 g/day of naringin for full responses, and 500 mg could be below the threshold for efficacy [43]. Additionally, naringin’s oral bioavailability in humans is limited; peak plasma concentrations after a typical grapefruit juice dose (providing ~135–200 mg naringin) are in the low micromolar range (0.3–1.5 μM) [66,67]. Inter-individual differences in gut metabolism of naringin to naringenin may also influence its systemic effects. These factors highlight the need for optimizing dosing in future trials. Nonetheless, the positive randomized controlled trial by Barajas-Vega et al. demonstrates that at an appropriate dose and duration, naringin can beneficially modify cardiovascular risk markers in humans, particularly related to dyslipidemia and adiposity [64].

#### 3.4.2. Clinical Studies on Vascular Function

The direct impact of naringin on human vascular function has been explored in the context of dietary flavonoid interventions. Grapefruit juice (GFJ) is rich in naringin (the major grapefruit flavanone is naringin, which is metabolized to naringenin in vivo). A 2015 randomized controlled crossover trial by Habauzit et al. tested whether chronic grapefruit juice consumption improves vascular function in healthy postmenopausal women [63]. In this study, 48 women consumed 340 mL of grapefruit juice daily (providing ~210 mg of naringin glycosides) for 6 months, and for a separate period, consumed a placebo drink without flavanones, with a washout in between. The primary endpoint was endothelial function measured by brachial artery flow-mediated dilation (FMD). Secondary endpoints included arterial stiffness (pulse wave velocity) and blood pressure. The results were somewhat mixed: central arterial stiffness significantly improved after 6 months of grapefruit juice compared to placebo. Specifically, carotid-femoral pulse wave velocity (a measure of aortic stiffness) was lower on grapefruit juice (7.36 m/s) than on placebo (7.70 m/s, *p* = 0.019). This indicates increased arterial elasticity, which is a favorable outcome associated with reduced cardiovascular risk. By contrast, endothelial function (FMD) did not significantly change with grapefruit juice in this study; FMD remained similar before and after, and not different from the control beverage. Additionally, no significant changes in peripheral endothelial function (assessed by digital pulse amplitude) or blood pressure were observed. The improvement in arterial stiffness without a measurable change in FMD suggests that grapefruit’s flavanones (like naringin/naringenin) may exert subtler effects on the vasculature that are not captured by brachial FMD, or that the baseline endothelial function of these healthy participants was normal and thus hard to improve. The authors noted that the arterial stiffness benefit might be related to flavanone-mediated improvements in vascular wall structure or function, possibly through enhanced nitric oxide (NO) bioavailability or reduced arterial wall inflammation. While FMD was unchanged, the reduction in pulse wave velocity is still a clinically relevant outcome, as lower central stiffness can reduce cardiac afterload and improve coronary perfusion. This trial provides human evidence that long-term consumption of naringin-rich foods (like grapefruit) can be beneficial for vascular health, even if the effects on classic endothelial function parameters are modest. It aligns with the notion that dietary flavonoids help maintain vascular elasticity in aging populations.

Another angle of human evidence comes from grapefruit consumption studies on blood pressure and atherogenic markers. Some small studies have reported that including grapefruit in the diet can lead to mild reductions in blood pressure and improvements in serum lipids, which have been partly attributed to naringin. [62]. For example, a pilot study observed that hypertensive patients who ate grapefruit daily for several weeks had a slight decrease in blood pressure and C-reactive protein levels compared to controls (though comprehensive controlled data are limited) [60]. Grapefruit’s high naringin content (and its active metabolite naringenin) is hypothesized to contribute to these effects via antioxidant and vasodilatory mechanisms, as supported by the animal studies on blood pressure. Clinical trial data on the potential therapeutic use of naringin has been recently summarized by Salehi et al. [7].

Overall, human data, while not yet extensive, suggest that naringin or naringin-rich foods can improve certain cardiovascular risk factors. The main challenge for clinical translation is ensuring adequate bioavailability and appropriate dosing to mirror the effective exposures seen in animal models. Future clinical trials with larger sample sizes, optimized naringin formulations, and direct measures of endothelial and cardiac function (e.g., FMD, echocardiography, exercise capacity) will be valuable to conclusively determine naringin’s cardioprotective potential in humans.

## 4. Mechanisms of Action and Discussion

A summary of Naringin’s cardioprotective effects are attributed to multiple converging mechanisms (Figure 4). Naringin’s cardiovascular effects are underpinned by multiple molecular mechanisms. Research from cell culture, animal models, and biochemical assays converge on the idea that naringin acts through a network of pathways that combat oxidative stress, inflammation, and cell death while promoting nitric oxide signaling and metabolic homeostasis. The following are key mechanisms by which naringin exerts its protective actions on the endothelium and myocardium.

### 4.1. Antioxidant Activity and Nrf2 Activation

Naringin is a potent antioxidant that directly scavenges reactive oxygen species and enhances the endogenous antioxidant defense system. Chemically, naringin’s polyphenolic structure can neutralize free radicals, as evidenced by in vitro assays (e.g., DPPH radical scavenging) [68]. In biological systems, naringin consistently reduces markers of oxidative damage. For instance, in hypercholesterolemic rat arteries, naringin lowered levels of superoxide anion (O_2_^−^) in the aortic wall, as visualized by reduced dihydroethidium (DHE) fluorescence signal after treatment [42]. It also decreased protein adducts of lipid peroxidation (4-HNE) and nitro-oxidative stress (3-nitrotyrosine) in those arteries [42]. In myocardial I/R models, naringin-treated hearts showed significantly lower MDA (a lipid peroxidation byproduct) and higher SOD activity than untreated hearts, indicating that naringin preserved antioxidant enzyme function during oxidative stress [40]. One likely mechanism for these effects is activation of the Nrf2 pathway, the master regulator of antioxidant genes. While naringin’s direct influence on Nrf2 in vivo is still being elucidated, related studies suggest it does engage this pathway [49,69]. Chen et al. reported that naringin promoted Nrf2 nuclear translocation in cardiomyocytes subjected to anoxia/reoxygenation, resulting in upregulation of Nrf2-dependent genes. By activating Nrf2, naringin can induce expression of phase II antioxidant enzymes (like glutathione peroxidase, heme oxygenase-1, catalase, and SOD), thereby augmenting the cellular capacity to neutralize ROS [69]. A study in diabetic rats found that naringin increased myocardial glutathione/glutathione disulfide (GSH/GSSG) ratio and activities of catalase and SOD compared to untreated diabetics, consistent with Nrf2 pathway activation (Heidary Moghaddam et al., 2020) [14]. Thus, naringin’s antioxidative mechanism involves both direct radical scavenging and indirect boosting of antioxidant defenses via Nrf2.

### 4.2. Inhibition of Inflammatory Signaling (NF-κB and Cytokines)

Naringin exerts strong anti-inflammatory effects in cardiovascular tissues, largely by suppressing the NF-κB signaling cascade and downstream proinflammatory mediators. In endothelial cells, naringin prevented the nuclear translocation of NF-κB-p65 by blocking upstream IKK and IκB-α phosphorylation in response to TNF-α [17]. This resulted in lower expression of NF-κB target genes encoding adhesion molecules (VCAM-1, ICAM-1, E-selectin) and chemokines (CX3CL1, MCP-1, RANTES)––molecules that orchestrate endothelial inflammation and leukocyte recruitment in atherosclerosis. By keeping NF-κB inactive in the cytosol, naringin effectively shuts off the inflammatory adhesion cascade in endothelial cells, which can reduce the initiation of atheroma formation. Similarly, in vascular smooth muscle cells, naringin inhibited NF-κB-dependent MMP-9 expression, which may stabilize plaques and reduce vascular inflammation [33]. In macrophages (though not detailed here), citrus flavonoids like naringin are known to curb inflammatory cytokine release (e.g., lowering TNF-α and IL-6 production) via NF-κβ and AP-1 pathway modulation [70,71].

In vivo, the anti-inflammatory action of naringin is evident from reduced cytokine levels in blood and tissue. Naringin-treated hypertensive rats had significantly lower plasma TNF-α than untreated hypertensive controls [1]. In myocardial ischemia reperfusion (I/R) rats, naringin blunted the reperfusion-induced spike in cardiac pro-inflammatory cytokines: levels of TNF-α, IL-1β, and IL-6 in heart tissue were all significantly suppressed with naringin pretreatment compared to I/R alone [47]. Such cytokine reduction is likely a consequence of upstream NF-κB inhibition, as well as reduced activation of inflammasomes (some studies suggest naringin/naringenin can inhibit NLRP3 inflammasome activation, thereby lowering IL-1β) [4]. Furthermore, by scavenging ROS, naringin diminishes redox-sensitive inflammatory pathways (ROS are potent activators of NF-κB and other inflammatory kinases). In a rat model of endotoxemia (sepsis), naringin was observed to protect cardiac function by attenuating NF-κB–mediated inflammation in the heart, again underscoring its anti-inflammatory capability in cardiovascular contexts [55,56]. In summary, naringin mitigates vascular and myocardial inflammation by impeding central inflammatory signals (like NF-κB) and reducing the release of damaging cytokines and chemokines.

### 4.3. Improvement of Endothelial Function and NO Bioavailability

A critical aspect of endothelial function is the production and bioavailability of nitric oxide (NO), which mediates vasodilation and has anti-atherogenic effects. Naringin positively influences NO pathways in the endothelium. As described in animal studies, naringin restored acetylcholine (ACh)-induced endothelium-dependent relaxation in hypercholesterolemic rat aortas [42]. This implies that naringin either increased endothelial NO synthesis (for example, by upregulating endothelial nitric oxide synthase, eNOS) or preserved existing NO from oxidative destruction. The evidence favors the latter mechanism; naringin dramatically lowered arterial superoxide levels and the expression of NADPH oxidase isoforms (major sources of superoxide). By reducing superoxide, naringin prevents NO from being rapidly inactivated (superoxide reacts with NO to form peroxynitrite). In essence, naringin enhances NO bioavailability by creating a less oxidative environment in the endothelium. Additionally, the downregulation of LOX-1 by naringin is relevant here: LOX-1 activation by oxidized LDL in endothelial cells not only triggers inflammation but also reduces eNOS activity and NO production. Naringin’s suppression of LOX-1 likely helps sustain normal eNOS function and NO release. Crucially, recent studies show that naringin directly influences eNOS activation via intracellular signaling. Specifically, naringin activates the PI3K/Akt pathway, a key regulator of endothelial NO production. Activation of PI3K leads to Akt phosphorylation, which in turn phosphorylates eNOS at Ser1177, a critical modification that increases eNOS activity and NO output [72,73]. In fact, the flavonoid’s activation of PI3K/Akt/eNOS signaling has been mechanistically linked to enhanced NO production and endothelial cell function under oxidative stress and diabetic-like conditions.

Some studies also indicate that citrus flavonoids, including naringin, increase eNOS expression and phosphorylation through PI3K/Akt, thereby promoting NO generation [74]. Thus, naringin not only preserves existing NO but may also upregulate eNOS enzymatic activity through Akt-mediated phosphorylation, offering a dual mechanism for enhancing endothelial NO signaling. The functional outcomes––improved vasodilation and lower blood pressure––suggest that naringin rescues endothelial NO signaling through both oxidative stress reduction and direct enzymatic activation pathways. The improved endothelial function in naringin-treated animals (e.g., better ACh response and lower vascular resistance) can be largely attributed to higher NO availability and reduced endothelial oxidative stress [42]. The vasorelaxant effect of flavonoids on isolated vascular smooth muscle has been shown to be concentration dependent and in large part NO driven [75]. This mechanism is crucial in explaining naringin’s protective effect against endothelial dysfunction in atherosclerosis and hypertension.

### 4.4. Modulation of the Renin-Angiotensin System (RAS)

Naringin appears to interfere with the RAS, which is often overactive in hypertension and heart failure. The L-NAME hypertensive rat study provided clear evidence of RAS modulation: naringin prevented the elevation of circulating Angiotensin II and down-regulated Ang II type 1 receptor (AT1R) expression in cardiac tissue [4]. It also normalized the balance of Ang II/ACE and ACE2 observed in hypertensive animals, suggesting a complex regulatory effect on angiotensin-converting enzyme (ACE) pathways [76]. In another study, naringin lowered renal ACE and mineralocorticoid receptor expression in hypertensive rats, contributing to blood pressure reduction [77]. Naringin’s suppression of AT1R signaling in the heart (and possibly vessels) leads to less vasoconstriction, lower oxidative stress (Ang II-AT1R signaling activates NADPH oxidase), and reduced inflammatory cascades. In the hypertensive rats, naringin also attenuated the upregulation of AT1R downstream effectors like protein kinase C (PKC) and the MAPK/ERK pathway in the left ventricle [4]. By inhibiting AT1R/PKC/ERK signaling, naringin likely prevented Ang II-induced cardiac hypertrophy and fibrosis in those animals. This mechanism aligns with in vitro data: in Ang II-stimulated cardiomyocytes, naringin blocked hypertrophic responses (cell enlargement and fetal gene expression) at least partly by reducing Ang II’s effective signaling (through CA-II/NHE1 inhibition, as discussed) and perhaps through directly dampening AT1R pathways. Moreover, naringin’s ability to increase Akt may counteract Ang II’s effects, since Akt activation can oppose hypertrophic signaling and promote NO release (Ang II often suppresses Akt/eNOS in endothelium; naringin might restore it). Therefore, naringin’s cardioprotection in hypertensive settings is linked to RAS inhibition, resulting in lower vasopressor activity and mitigation of Ang II-mediated end-organ damage. This RAS-modulatory effect is an important facet of naringin’s mechanism, especially relevant to blood pressure control and prevention of pathological cardiac remodeling.

### 4.5. Anti-Apoptotic and Cell Survival Pathways

Naringin helps cells survive stress by modulating apoptosis-regulating proteins and signaling pathways. In ischemic myocardium, naringin consistently increases the Bcl-2/Bax ratio (favoring cell survival) and reduces the activation of executioner caspases. For example, naringin-pretreated I/R hearts showed higher Bcl-2 levels and lower cleaved caspase-3 levels than untreated hearts [47]. In vitro, naringin protected H9c2 cardiomyocytes from hypoxia-induced apoptosis by preserving mitochondrial membrane potential and activating pro-survival kinases. A central player here is the PI3K/Akt pathway, which, when activated by naringin, phosphorylates downstream targets that inhibit apoptosis (such as Bad and caspase-9) and enhance survival (like CREB and eNOS). Li et al. (2021) explicitly demonstrated that naringin’s activation of Akt led to reduced apoptosis in I/R, whereas blocking Akt partly restored the high apoptosis levels despite naringin [47,78]. Additionally, naringin’s anti-apoptotic effect is intertwined with its antioxidant effect: by reducing oxidative DNA damage and mitochondrial dysfunction, naringin prevents the activation of intrinsic apoptotic pathways. In doxorubicin cardiotoxicity, for instance, naringin decreased cardiomyocyte apoptosis by suppressing p38 MAPK and increasing endogenous antioxidants (doxorubicin causes apoptosis via oxidative stress and p38 activation, which naringin countered) [31,59]. Overall, by modulating key checkpoints of apoptosis (Bcl-2 family, caspases, survival kinases), naringin helps endothelial and cardiac cells resist programmed cell death during stress, contributing to tissue preservation.

### 4.6. Inhibition of Autophagy and Ferroptosis (Stress-Induced Cell Death)

Beyond apoptosis, cells under stress can undergo other forms of programmed death, such as autophagy-associated cell death and ferroptosis (iron-dependent lipid peroxidation-driven death). Naringin has shown the capacity to modulate these processes. In conditions of metabolic or oxidative stress, excessive autophagy can deplete cells of essential components, leading to cell death. A study found that in endothelial cells exposed to high glucose and fat (simulating diabetic/hyperlipidemic stress), naringin inhibited overactive autophagy via the PI3K-Akt-mTOR pathway, thereby ameliorating endothelial dysfunction [72]. Normally, high glucose/fat stress can trigger autophagy and also cause endothelial dysfunction; by activating Akt and mTOR, naringin signaled the cells to reduce autophagic flux, which preserved endothelial cell integrity and function. Similarly, in the heart I/R model, as noted, naringin lowered the levels of autophagy markers (Beclin-1 and LC3-II) that were elevated by ischemic stress [47]. This suggests naringin prevents ischemia-induced autophagosome formation, which might be harmful in excess. Regarding ferroptosis, a novel form of cell death caused by iron-catalyzed lipid peroxidation, naringin’s antioxidant role is critical. The study by Zhang et al. showed naringin reduced ferroptosis in I/R by inhibiting the cyclic GMP-AMP synthase, which stimulates the interferon genes (cGAS-STING) pathway, which otherwise promotes inflammation and possibly ferroptotic damage [30]. By preserving mitochondrial function (mitochondrial dysfunction is a hallmark of ferroptosis initiation) and reducing ROS, naringin protected cells from lipid peroxidation and the iron-dependent demise characteristic of ferroptosis. Naringenin (the aglycone of naringin) has also been reported to activate Nrf2/GPX4, a major antiferroptotic axis, in cardiomyocytes, and it is plausible that naringin, once converted to naringenin in vivo, contributes similarly to upregulating glutathione peroxidase 4 (GPX4) and other ferroptosis defense systems [49,79]. In summary, naringin modulates non-apoptotic cell death pathways. It can restrain maladaptive autophagy and protect against ferroptosis, which further contributes to cell survival under chronic stress or acute I/R injury.

### 4.7. Metabolic Enzyme Modulation and Improved Energy Utilization

Though not directly an “endothelial or myocardial function” mechanism, naringin’s metabolic effects are worth noting, as they indirectly benefit cardiovascular health. Naringin has been shown to activate AMP-activated protein kinase (AMPK) in metabolic tissues [80]. Activation of AMPK leads to improved energy balance, increased fatty acid oxidation, and reduced lipid accumulation, which can alleviate lipotoxicity in the heart and vessels. In high-fat-fed mice, naringin activated AMPK in skeletal muscle and liver, contributing to reduced plasma lipids and prevention of insulin resistance [80]. It also upregulated the transcription factor peroxisome proliferator-activated receptor gamma (PPARγ) and downregulated lipogenic enzymes (like fatty acid synthase and HMG-CoA reductase) in rodent models, thereby improving dyslipidemia and possibly adipose tissue inflammation [81]. By lowering circulating cholesterol and triglycerides, naringin reduces substrate for atherosclerosis. Additionally, its ability to inhibit dipeptidyl peptidase 4 (DPP-4) enzyme (one study noted a DPP-4 inhibitory effect at 40 mg/kg in rats) can increase incretin levels and improve glycemic control, indirectly benefiting endothelial function through reduced glucotoxicity [82]. Improved mitochondrial function is another mechanism: naringin-treated obese rats had enhanced liver mitochondrial respiratory function and efficiency [43]. Additionally, attenuation of mitochondrial dysfunction in glucose induced cardiomyocyte apoptosis via modulation of the p38 signaling pathway has also been shown with naringin treatment [83]. Efficient mitochondrial function in muscle and heart means less oxidative stress and better ATP supply, which can protect against heart failure and endothelial energy deprivation. Indeed, in diabetic hearts, naringin improved mitochondrial complex activity and reduced mitochondrial ROS production (a likely consequence of AMPK/Nrf2 activation). Overall, naringin’s metabolic reprogramming (enhancing AMPK, PPARγ, and reducing lipid accumulation) creates a systemic environment that is less atherogenic and more supportive of cardiovascular health.

In summary, naringin operates through a constellation of interrelated mechanisms to confer cardiovascular protection. It is a multifaceted agent: an antioxidant and anti-inflammatory molecule that preserves NO and endothelial function; an inhibitor of pathological signaling (NF-κβ, AT1R, PKC, MAPKs) that drives vascular and cardiac remodeling; an activator of cell-survival pathways (PI3K/Akt, Nrf2) that prevent cell death; and a modulator of metabolism that reduces risk factors like dyslipidemia and hyperglycemia. These mechanisms are highly synergistic. For example, by reducing oxidative stress, naringin indirectly improves NO signaling and reduces NF-κB activation (since oxidative stress activates NF-κβ), and by activating Akt, it not only blocks apoptosis but also increases eNOS and Nrf2 activity, compounding its antioxidant and pro-endothelial effects. This network of actions enables naringin to target the root causes of cardiovascular dysfunction––oxidative damage and inflammation––as well as the downstream consequences––endothelial dysfunction, hypertension, atherosclerosis progression, and myocardial injury. In addition to the above mechanisms, a recent study on peripheral blood mononuclear cells of participants of a clinical trial on the effects of grapefruit juice flavanones (6 months) in post-menopausal women showed that consumption of flavanones induced changes in the gene expression profile of protein coding genes and non-coding microRNA genes. These genes are part of the complex network that regulates inflammation, cell adhesion, and mobility. The authors concluded that regular consumption of flavanones could modulate the expression of genes that contribute to prevention of vascular dysfunction and development of cardiovascular disease [84,85]. It is this broad spectrum of molecular and genomic influence that underlies the significant improvements observed in experimental models and hints at therapeutic potential in humans.

## 5. Conclusions

In conclusion, the body of evidence from the past two decades positions naringin as a potent natural compound with cardiovascular benefits. Its ability to improve endothelial function and protect the myocardium in the face of diverse insults––from high-fat diets and hypertension to acute ischemic injury––is well-documented in preclinical models. Naringin emerges as a multi-targeted agent: it suppresses oxidative stress, quells inflammation, preserves nitric oxide and endothelial integrity, modulates neurohormonal systems and activates pro-survival signaling in cells while inhibiting apoptosis, autophagy, and ferroptosis.

This pleiotropic activity is particularly beneficial in complex cardiovascular conditions such as atherosclerosis and heart failure, where numerous pathological mechanisms are activated concurrently. By simultaneously modulating multiple pathways, naringin may exert a more robust cardioprotective effect than agents targeting a single mechanism. Moreover, emerging evidence of naringin’s role in epigenetic regulation, including modulation of non-coding RNAs and chromatin remodeling, adds an important dimension to its therapeutic potential [85]. While cautious optimism is warranted until more clinical data accumulate, naringin represents a compelling example of a nutraceutical that could complement traditional cardiovascular drugs. Importantly, naringin appears to be safe and well-tolerated in humans at the doses studied (up to ~500 mg/day in supplements or the equivalent from diet) [61]. No serious adverse effects were reported in the trials aside from expected grapefruit–drug interaction considerations. This safety profile, combined with its natural occurrence in common fruits, makes naringin an attractive candidate for preventive nutrition or adjunct therapy in cardiovascular disease.

Despite the positive findings, it must be acknowledged that further research is needed to cement naringin’s role in clinical practice. Large-scale randomized trials in patients (e.g., those with coronary artery disease, heart failure, or hypertension) are necessary to determine whether naringin supplementation can improve “hard” outcomes like reduction in cardiac events, improvement in exercise tolerance, or regression of atherosclerosis. Optimal dosing and bioavailability enhancement strategies (such as using naringin nanocarriers or co-administering with absorption enhancers) should be investigated. Additionally, research should explore population subsets––for example, whether individuals with certain genotypes or gut microbiome compositions respond differently to naringin (given that conversion to naringenin by gut flora is a key step) [86]. Mechanistic studies in humans (perhaps using vascular imaging or myocardial strain analysis) could help confirm that the vascular improvements seen experimentally (like enhanced endothelial function and reduced inflammation) occur in patients taking naringin.

The integration of naringin, whether through dietary intake or targeted supplementation, may represent a complementary strategy for the prevention and management of cardiovascular disease. By enhancing endothelial function, attenuating key risk factors, and mitigating myocardial injury, naringin demonstrates multifaceted cardioprotective potential. This review also highlights a substantial translational gap between the extensive preclinical evidence from cellular and animal studies and the limited number of human clinical trials. Investigations into the pharmacokinetics and bioavailability of various naringin formulations––particularly those capable of achieving plasma concentrations comparable to the high doses used in animal models––are critically needed. Future cardiovascular clinical trials should incorporate long-term follow-up of cardiovascular endpoints and evaluate naringin’s effects in conjunction with established pharmacotherapies. Additionally, the immunomodulatory role of naringin in cardiovascular disease remains poorly understood, particularly its interaction with adaptive immune cells such as T cells and macrophages in the context of atherosclerosis.

Considering the global burden of CVD and the pressing need for accessible, low-risk therapeutic options, further clinical investigation into the efficacy and safety of naringin is warranted. The evidence synthesized in this review provides a compelling rationale for such studies and underscores the potential of this citrus-derived flavonoid as a supportive agent in cardiovascular health promotion.

## Figures and Tables

**Figure 1 nutrients-17-02658-f001:**
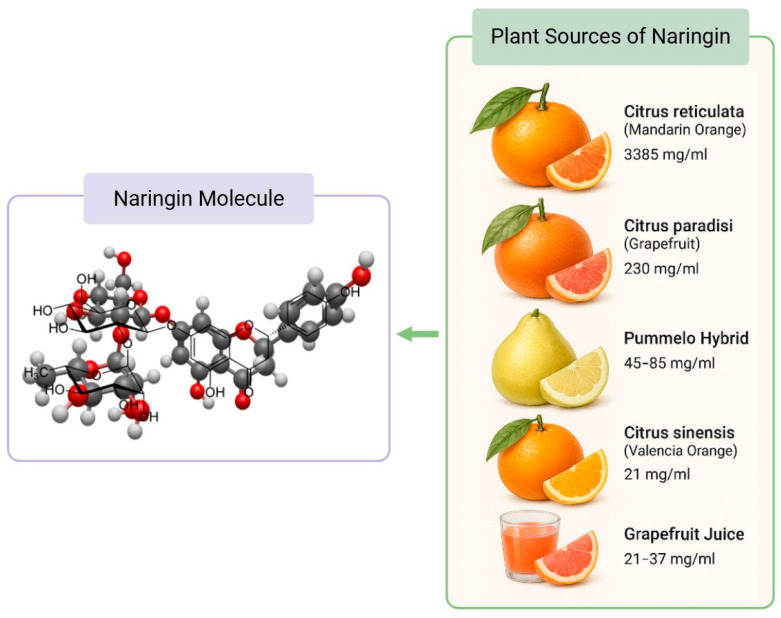
Naringin Molecular Structure and Citrus Food Sources. The concentration of naringin in plant sources were obtained from Alam et al. [3].

**Figure 2 nutrients-17-02658-f002:**
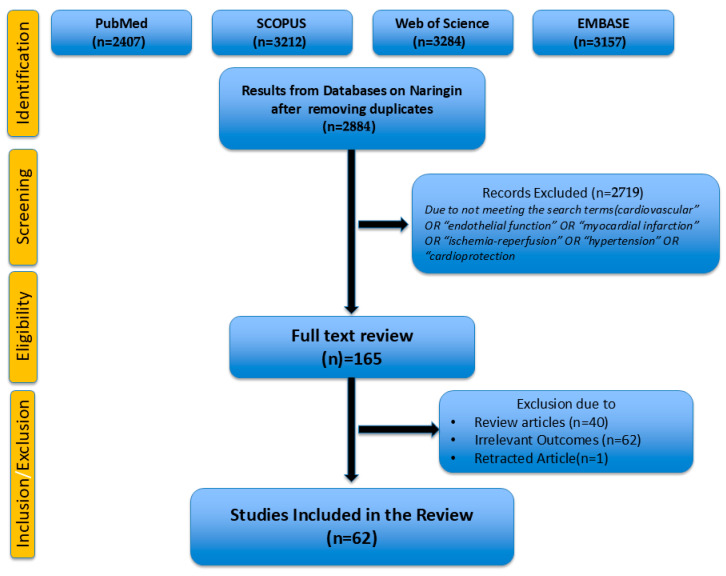
PRISMA Flow Diagram.

**Figure 3 nutrients-17-02658-f003:**
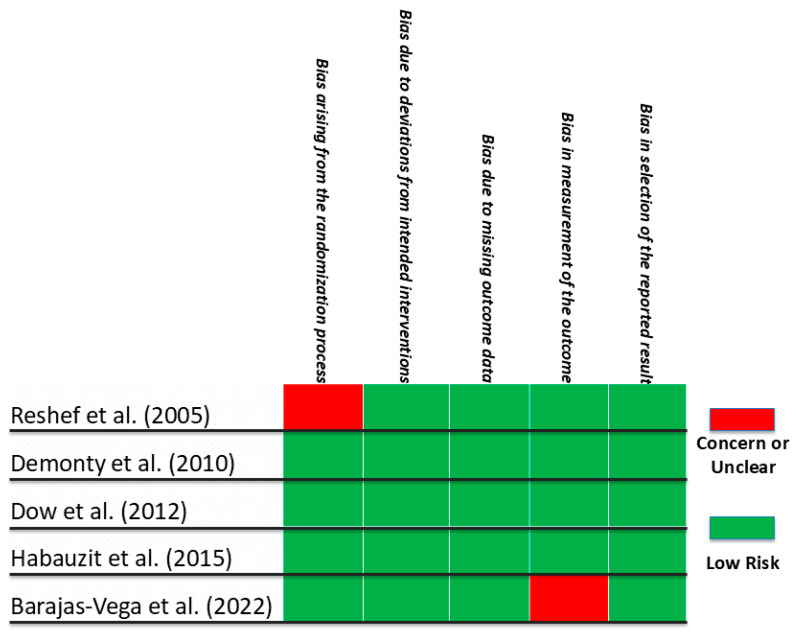
Risk of Bias 2 (RoB2) for Human Clinical Trials from January 2000 to June 2025 on the Effects of Naringin on Cardiovascular Endpoints Reshef et al. [60], Demonty et al. [61], Dow et al. [62], Habauzit et al. [63], Barajas-Vega et al. [64]. Areas shaded in red represent a concern or unclear risk of bias. Areas shaded in green represent a low risk of bias.

**Figure 4 nutrients-17-02658-f004:**
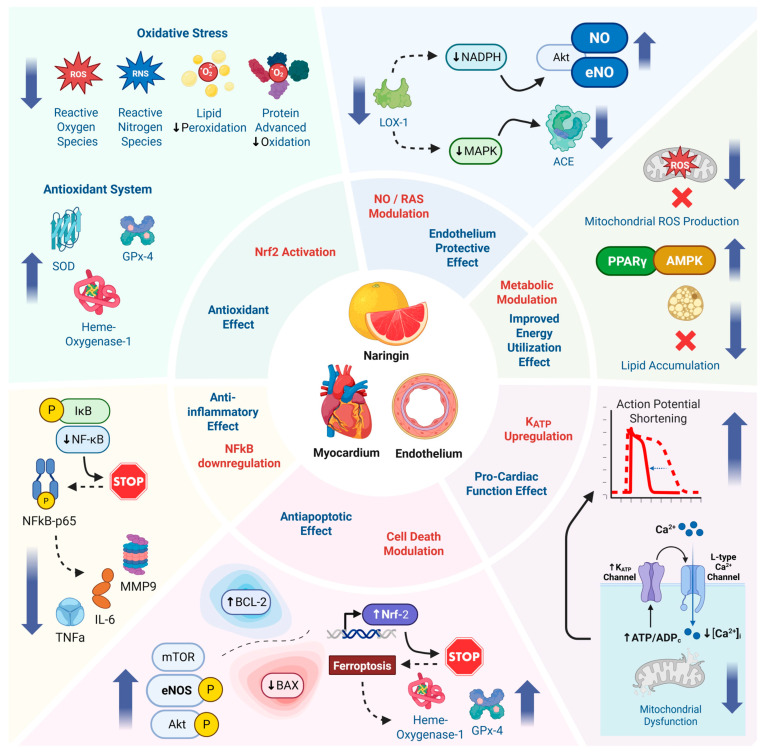
Schematic representation of the proposed cardioprotective mechanisms of naringin, highlighting its antioxidant, anti-inflammatory, anti-apoptotic, endothelial-protective, and metabolic effects via key pathways in endothelial and myocardial tissues.

**Table 1 nutrients-17-02658-t001:** Human Clinical Trials from January 2000 to June 2025 on the Effects of Naringin on Cardiovascular Endpoints.

Reference and Year	Study Type	Population Studied	Mean Age ± SD	Dosage of Naringin Used	Duration of Treatment	Key Outcomes
Reshef et al. (2005) [60]	Double Blind Cross Over	12 adults with Stage I hypertension	52.1 ± 10.1	0.5 L/day of high-flavonoid sweetie juice (677 mg/L naringin)	5 weeks (cross-over design)	Reduced diastolic BP significantly with HF juice vs. LF juice; systolic BP not significantly different
Demonty et al. (2010) [61]	Double Blind Placebo Controlled	204 moderately hypercholesterolemic men and women (n = 64 Naringin)	59.8 ± 8.8	500 mg/day (capsules)	4 weeks	No effect on TC, LDL, HDL or triglycerides compared to placebo
Dow et al. (2012) [62]	Randomized Control	74 overweight adults (n = 39 Naringin)	39.4 ± 10.7	1.5 grapefruits/day (fresh Rio-Red grapefruit)	6 weeks	Reduced waist circumference and systolic BP; modest weight loss
Habauzit et al. (2015) [63]	Randomized Crossover Control	48 healthy postmenopausal women	50–65 years; Mean ± SD not explicitly stated	340 mL/day grapefruit juice (~210 mg naringenin glycosides)	6 months (cross-over design with 2-month washout)	Reduced carotid-femoral pulse wave velocity; no effect on BP or other metabolic markers
Barajas-Vega et al. (2022) [64]	Double Blind Randomized	28 adults with dyslipidemia and Class I obesity (n = 14 Naringin)	Naringin group: 50.1 ± 5.48 years	450 mg/day (capsule, 98% purity)	90 days	Reduced weight, BMI, total and LDL cholesterol; increased adiponectin

Key outcomes: BP (blood pressure), HF (High Flavonoid), LF(Low Flavonoid), TC (Total Cholesterol), LDL (Low Density Lipoprotein), HDL (High Density Lipoprotein), BMI (Body Mass Index).

## Data Availability

Data are contained within the article and Appendix A.

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
