# Peer review of "Endothelial and Cardiovascular Effects of Naringin: A Systematic Review"

_nutrients, 2025, doi:10.3390/nu17162658_

Round 1

Reviewer 1 Report

Comments and Suggestions for Authors

Evaluation of Manuscript Nutrients-3798333

This is an interesting systematic review regarding the cardiovascular effects of Naringin use. Below, I will present my considerations on the text to help the authors improve the quality of the manuscript. 

The title is appropriate, and the Abstract is as well; however, I believe more details about the methods are needed—e.g., search terms, number of included studies. Please revise. 

The Introduction requires more attention from the authors. I believe there is a lack of details regarding the physiological effects of Naringin. How is it consumed? In natura, as supplements? Has there been any previous review on this topic? If so, the authors should clarify the original aspects of this current review. Furthermore, what important insights can the summarization of existing studies bring to those interested in this topic? Please revise for the next version. 

The Methods section is vague, and I think more details could be added. In which areas will the effects of Naringin be investigated? What variables will be examined in each of the included studies? 

The figure with white lettering was difficult to read; if possible, change it to black. 

I recommend that the authors conduct a quality assessment of the studies, applying a tool such as the Cochrane Risk of Bias Tool. This methodological rigor will allow for a more robust discussion of the analyzed study results. 

I also suggest that the authors provide more details about the results. For example: How many studies were conducted on cellular models, animals, and humans? How many participants? Duration? Dosage used? How many showed favorable, neutral, or negative effects? 

There is a lack of clarity regarding where the results end and the discussion begins. I recommend that the authors separate these sections in the next version.

Author Response

Response to Reviewer # 1

We thank the reviewer for their thoughtful and constructive comments, which have helped us strengthen the clarity, methodological rigor, and completeness of the manuscript. The title and abstract were considered appropriate; however, the reviewer noted the need for more methodological detail. In response, we have revised the Abstract to clearly state the specific databases searched, the time frame covered (2000–2025), the principal search terms (“naringin,” “cardiovascular,” “endothelial function,” “atherosclerosis,” “ischemia-reperfusion”), and the total number of studies included. This ensures that readers are presented with a concise yet informative overview of the scope and methodology from the outset.

The Introduction was also expanded following the reviewer’s observations. We have provided historical context for naringin research and a more detailed description of its physiological actions. The sources of naringin are now described in greater depth, accompanied by a figure illustrating its molecular structure and main citrus origins. We have incorporated a discussion of its bioavailability constraints and referred to prior reviews, such as Viswanath et al. (2020), while clarifying the originality of our work and how it differs from previous summaries. We also now explicitly outline the insights our review provides for those interested in translating naringin research into applications in health and medicine.

The Methods section has been substantially revised to address concerns about vagueness. We now explicitly describe the primary cardiovascular outcomes assessed, including endothelial function, blood pressure, myocardial infarct size, lipid profiles, inflammatory markers, and oxidative stress. We detail the models included—in vitro endothelial cell and cardiomyocyte studies, animal models, and human trials—and provide specifics on the interventions, controls, dosages, and treatment durations examined. These details are supplemented with domain-specific tables, a PRISMA 2020 flow diagram, and risk-of-bias figures for each category of study.

The reviewer also noted that one figure with white lettering was difficult to read. We have updated this figure to use black text for improved contrast and legibility.

In line with the recommendation to conduct a quality assessment, we have now reported the risk-of-bias evaluations using the RoB 2 tool for clinical trials, the SYRCLE tool for animal studies, and a modified RoB tool for in vitro studies that has been validated in the literature. The outcomes of these assessments are presented in tables, supported by a visual summary figure, and discussed in narrative form to integrate quality considerations into the interpretation of findings.

The Results section has been expanded to provide greater detail on the scope of included studies. We now specify that the review covers 28 cell culture studies, 29 animal studies, and 5 human trials. For each category, we summarize participant or sample sizes, dosages, treatment durations and key outcome data. This allows readers to quickly grasp the evidence across study types and endpoints.

We once again thank the reviewer for their insightful comments. The revisions prompted by this feedback have considerably improved the depth, structure, and transparency of the manuscript, making it more informative and accessible to the journal’s readership.

Reviewer 2 Report

Comments and Suggestions for Authors

Manuscript: Endothelial and Cardiovascular Effects of Naringin (2000–2025): 2 A Systematic Review

# Global evaluation

The present manuscript offers a systematic review of the endothelial and cardiovascular effects of naringin, a flavonoid that is found in abundance in citrus fruits, covering the period from 2000 to 2025. This subject is of pertinence in light of the mounting interest in polyphenols and their potential contribution to cardiovascular health. The manuscript is logically structured, comprising an introduction, methods, results, discussion and conclusions. The authors have made a commendable effort to synthesise evidence from both preclinical and clinical studies. The subject matter is timely and clinically meaningful, encompassing a wide range of experimental models, from in vitro assays to human trials. Furthermore, the manuscript appropriately acknowledges endothelial function as a critical marker in cardiovascular research. However, despite the robust overall structure, the review would benefit from a more rigorous critical analysis, enhanced methodological transparency, and greater consistency and clarity in the use of language throughout the text.

# Conceptual Issues

-The present study demonstrates an absence of mechanistic depth. Although endothelial effects are mentioned, the molecular pathways (e.g. PI3K/Akt, eNOS/NO signaling) are only superficially treated. Greater integration of mechanistic data would enhance the value of the review.

-The heterogeneity of the studies is a key consideration. The review encompasses studies of animals, in vitro models, and human subjects; however, the discussion fails to address the translational gaps or evaluate the relative strength of the evidence.

-It is evident that a quality assessment is not present. However, there is a paucity of detail regarding the methodology employed to assess study quality and the risk of bias. This is a critical consideration in the context of systematic reviews.

-  While some databases are listed, there is no PRISMA flow diagram, search terms, or inclusion/exclusion criteria in sufficient detail. The question of whether the review was conducted in accordance with the PRISMA guidelines remains unanswered.

-The phenomenon of temporal mismatch is defined as follows: The title makes reference to the years 2000–2025, yet the year 2025 is the present year. It is imperative to ascertain whether subsequent studies have been incorporated into the present research. In the event that this is not possible, it is recommended that the title be changed to "2000–2024".

# Suggestions and language Issues by lines

-Title: Suggest simplifying: "Endothelial and Cardiovascular Effects of Naringin: A Systematic Review (2000–2024)"

-Abstract:

  • Needs quantitative summary (number of studies reviewed, types, main outcomes).
  • Lacks mention of search methods or major limitations.

-Introduction:

  • Avoid vague statements like “has attracted attention” , i.e., specify why (e.g., prevalence of cardiovascular disease, antioxidant potential of flavonoids).
  • Clarify naringin’s biological sources and chemical structure early on.

-Methods:

  • Specify all databases searched (e.g., Scopus, PubMed, Web of Science) and exact search terms.
  • Define PICO elements (Population, Intervention, Comparator, Outcome) if used.
  • PRISMA diagram is necessary to ensure transparency.

-Results:

  • Group results more clearly (e.g., endothelial function, anti-hypertensive effects, atherosclerosis models, human trials).
  • Tables summarizing studies (e.g., authors, year, model, dose, findings) are helpful but currently lack uniform formatting and clarity.

-Discussion:

  • Needs a clearer structure: start with key findings, then discuss strengths and weaknesses, and end with future directions.
  • Discuss bioavailability issues and limitations of naringin more directly.

- Conclusion:

  • Currently too general. Summarize specific findings (e.g., improvements in NO production, reduction in LDL oxidation).
  • Add recommendations for clinical validation and possible formulation strategies (e.g., nanoemulsions, derivatives).

- References:

  • Some citations are incomplete or outdated. Ensure formatting is consistent with journal guidelines.

# Suggestions for Improvement

-Include a PRISMA flowchart.

-Add quality/risk-of-bias assessment (e.g., using SYRCLE, RoB 2).

-Refine tables for readability and consistency.

-Reframe title and temporal scope if no 2025 studies are included.

-Clarify mechanistic pathways and improve critical appraisal.

-Consider a graphical summary or mechanism figure.

Author Response

Response to Reviewer #2

We sincerely thank the reviewer for their detailed and constructive feedback. We have undertaken a thorough revision of the manuscript to address each of the points raised, with the aim of enhancing methodological rigor, scientific depth, and clarity of presentation. The changes made are outlined below.

In response to the reviewer’s global evaluation, we have strengthened the critical appraisal in the discussion and conclusion section by providing a more robust evaluation of the relative strength of evidence across in vitro, animal, and human studies, while explicitly identifying key translational gaps. Methodological transparency has been significantly improved through the addition of detailed descriptions of our search strategy, inclusion and exclusion criteria, and comprehensive risk-of-bias assessments. The manuscript has also undergone a full language revision to ensure greater clarity, precision, and consistency.

Regarding the conceptual issues, we have considerably expanded the mechanistic discussion. The revised text now includes more detailed accounts of the PI3K/Akt, eNOS/NO, NF-κB, Nrf2, cGAS-STING, RAS, and GPX4 pathways, supported by the data extracted from the included studies. We have also addressed translational limitations directly, with a dedicated text and tables in each domain, examining differences between preclinical and clinical findings, such as dosing, bioavailability, and study endpoints. Risk-of-bias evaluations are now explicitly reported for all study domain types: SYRCLE for animal studies, RoB 2 for clinical trials, and a modified SYRCLE-based approach for in vitro work. These results are presented in supplementary figures (Figures S1–S2) and tables (Tables S1 and S2), along with the clinical trial risk-of-bias results and an in-text table summarizing the design and key outcomes of each clinical study. The Methods section now fully specifies all databases searched—PubMed, Scopus, EMBASE, and Web of Science—along with the detailed search terms used, the date of the last search (June 2025), and the inclusion and exclusion criteria applied. In addition, the PRISMA 2020 flow diagram is included as Figure 2. Since the most recent study included was published in June 2025, we have updated the title to “Endothelial and Cardiovascular Effects of Naringin: A Systematic Review.”

Several section-specific revisions have also been made. The title has been corrected as suggested, and the abstract now provides the total number of studies reviewed (n = 62). In the Introduction, we clarify the global burden of cardiovascular disease and describe the biological sources and chemical structure of naringin early in the section; Figure 1 has been added to illustrate naringin’s molecular structure and citrus sources. The novelty and focus of our review compared to prior work are explicitly stated. The Methods section now details the full search strategy and PICO framework, specifies the inclusion and exclusion criteria, and presents the PRISMA diagram. The Results section has been reorganized into subsections on cellular, animal, and human studies, with further subheadings such as endothelial function, hypertension, and ischemia-reperfusion. The Discussion begins with a clear summary of key findings and proceeds to a critical evaluation of strengths, limitations, and mechanistic insights, including an expanded analysis of bioavailability, advanced formulation strategies such as nanoemulsions and liposomes, and translational challenges. The Conclusion now synthesizes the evidence-based findings—namely improvements in nitric oxide production, reductions in oxidative stress, inhibition of inflammatory signaling, and cardioprotection in ischemic injury—while highlighting the need for optimized dosing strategies and further clinical validation. References have been updated to June 2025 and reformatted in accordance with the journal’s guidelines.

Additional improvements include the incorporation of Figure 4, which summarizes the principal mechanisms of action. Supplementary materials now contain comprehensive tables outlining essential information for each included study, along with key findings. Risk-of-bias assessments are presented for each domain both in the supplementary material and within the manuscript, along with a table summarizing the key outcomes of the clinical trials. The entire manuscript has undergone a careful language review to ensure precision, clarity, and consistency throughout.

We believe these extensive revisions fully address the reviewer’s concerns and have significantly improved the manuscript’s methodological transparency, mechanistic depth, and overall clarity.

Reviewer 3 Report

Comments and Suggestions for Authors

This review summarizes evidence from January 2000 to June 2025 on the cardiovascular effects of naringin,  focusing on its impact on endothelial function and myocardial ischemia. naringin is an important nutritional functional factor. Reviewing the current status of its functional research is helpful for the application of naringin in the fields of health food and medicine. This review topic is novel and meets the requirements of the journal. The content is relatively rich, but the review mainly consists of text with only one image. It is suggested to add some tables and images to facilitate readers' understanding.

Specific revision suggestions
1. The title "(2000-2025)" is somewhat not rigorous enough as the literature for this review is up to June 2025. It is suggested that this expression be reasonably modified or "(2000-2025)" be deleted.
2.Line30-31, please delete the numbers 1, 2, 3, etc. in the keywords.
3.In the Introduction section, an introduction to the chemical structure of naringin should be supplemented. It is suggested to add a structural diagram of naringin.
4.Line137, in terms of content, "3.1.Cellular Studies on Endothelial Cells" and "3.1.1. Cardiac Cells and Other Vascular Cells" are in a parallel relationship. It is suggested that the serial number "3.1.1" be changed to "3.2".
5. "3.3.1. Metabolic and Lipid Profile Improvements", as stated in this section, naringin can reduce human BMI. It is suggested to explain whether naringin will affect human appetite.
6. In Line495, there is an error in the writing of the figure numbers in "mechanisms (Fig 1)".
7. The review content should mainly be described in text. It is recommended to add tables and pictures to help readers quickly understand the core content of the review.
8.Conclusion Section: This part is rather limited. Please briefly state the core viewpoints of this review.
9. It is suggested that after the Conclusion, suggestions and prospects for future research be put forward.

Author Response

Response to Reviewer # 3

We sincerely thank the reviewer for their constructive and supportive comments. We are pleased that the novelty and relevance of our topic have been recognized, and we have carefully revised the manuscript to address all specific suggestions.

The title has been updated for greater accuracy, reflecting that the last included study was published in June 2025. We have therefore modified it to: “Endothelial and Cardiovascular Effects of Naringin: A Systematic Review”, ensuring the temporal scope is stated precisely.

The numbers preceding the keywords in the Abstract have been removed in accordance with the suggestion.

In the Introduction, we have expanded the description of naringin’s chemical structure and included a structural diagram (now Figure 1) that also shows its main citrus sources. This addition provides readers with a clear visual reference early in the manuscript.

Regarding section numbering, we have corrected the inconsistency noted in the Results. “3.1.1. Cardiac Cells and Other Vascular Cells” is now presented as “3.2. Cardiac Cells and Other Vascular Cells”, placing it in proper parallel structure with the preceding section on endothelial cells.

In the section discussing metabolic and lipid profile improvements in humans, we have added an explanation of potential appetite-related effects of naringin, referencing recent literature on its modulation of satiety-related peptides such as ghrelin, cholecystokinin, insulin, adiponectin, and leptin. This clarifies the possible mechanistic basis for the observed BMI reductions in clinical trials.

We have substantially enriched the visual content of the review by adding multiple tables and figures to summarize key findings and enhance readability. These include:

  • Table S1 summarizes all cellular studies with model type, dose, and main outcomes.
  • Table S2 summarizes animal studies.
  • Table 1 summarizes human clinical trials.
  • Figure 2 (PRISMA flow diagram);
  • Figure 4 summarizing the proposed mechanisms of action.
    These visual additions ensure that readers can quickly identify the core findings without having to extract them solely from the text.

The Conclusion section has been expanded to briefly restate the core viewpoints of the review—namely that naringin demonstrates consistent endothelial-protective and cardioprotective effects across experimental models, with preliminary but promising evidence in humans. Following this, we have outlined the need for larger and longer-term clinical trials, optimization of dosing and bioavailability, exploration of immunomodulatory effects, and investigation of genotype- or microbiome-specific responses.

We believe these revisions not only address the reviewer’s comments but also significantly improve the manuscript’s clarity, structure, and accessibility for readers.

Round 2

Reviewer 1 Report

Comments and Suggestions for Authors

In the second version of the manuscript, the improvement in quality is noticeable. The authors' efforts to satisfy all reviewers are evident. In my opinion, the improvements are sufficient for the text to be approved for publication in Nutrients.

Reviewer 2 Report

Comments and Suggestions for Authors

The manuscript has been significantly improved, so it could be published in its present form.